# Current Knowledge of Hepatitis E Virus (HEV) Epidemiology in Ruminants

**DOI:** 10.3390/pathogens11101124

**Published:** 2022-09-29

**Authors:** Federica Di Profio, Vittorio Sarchese, Andrea Palombieri, Paola Fruci, Gianvito Lanave, Serena Robetto, Vito Martella, Barbara Di Martino

**Affiliations:** 1Faculty of Veterinary Medicine, Università degli Studi di Teramo, 64100 Teramo, Italy; 2Department of Veterinary Medicine, Università Aldo Moro di Bari, 70121 Valenzano, Italy; 3Istituto Zooprofilattico Sperimentale del Piemonte, Liguria e Valle d’Aosta, CeRMAS, 11020 Aosta, Italy

**Keywords:** hepatitis E virus (HEV), epidemiology, domestic ruminants, wild ruminants, public health

## Abstract

Hepatitis E virus (HEV) infection represents an emerging public health concern worldwide. In industrialized countries, increasing numbers of autochthonous cases of human HEV infection are caused by zoonotic transmission of genotypes 3 and 4, mainly through the consumption of contaminated raw or undercooked meat of infected pigs and wild boars, which are considered the main reservoirs of HEV. However, in the last few years, accumulating evidence seems to indicate that several other animals, including different ruminant species, may harbor HEV. Understanding the impact of HEV infection in ruminants and identifying the risk factors affecting transmission among animals and to humans is critical in order to determine their role in the epidemiological cycle of HEV. In this review, we provide a summary of current knowledge on HEV ecology in ruminants. A growing body of evidence has revealed that these animal species may be potential important hosts of HEV, raising concerns about the possible implications for public health.

## 1. Introduction

Hepatitis E virus (HEV) is one of the most common causes of acute viral hepatitis worldwide [1]. HEV infection is underdiagnosed, since it is usually asymptomatic or self-limited in immunocompetent patients. However, HEV infection may progress to symptomatic acute liver failure in pregnant women or in individuals with pre-existing liver disease. Furthermore, the majority of HEV infections in immunosuppressed individuals can progress into chronicity which requires antiviral treatment, and chronic infection may lead to cirrhosis and transplantation [2]. Extra-hepatic manifestations have been also described, including neurological sequelae, such as Guillain–Barré syndrome and neuralgic amyotrophy, glomerulonephritis and pancreatitis [1]. 

HEV is a positive-sense, single-stranded RNA virus of about 6.4–7.2 kb in length. In the infected host, HEV exists in two forms: as non-enveloped virions (neHEVs) of 27–34 nm in diameter when secreted in faeces and bile, and as quasi-enveloped (eHEV) particles in circulating blood and in infected cell-culture supernatants [3,4]. The HEV genome, capped at the 5′ end with 7-methylguanosine and polyadenylated at the 3′ end, is organized in three partially overlapping open reading frames (ORFs), ORF1, ORF2, and ORF3, flanked by short 5′ and 3′ untranslated regions [5]. According to the 2021 release of the International Committee on the Taxonomy of Viruses (ICTV) [6], HEV is classified in the family *Hepeviridae*, consisting of two subfamilies, five genera and ten species. Members of the subfamily *Parahepevirinae* infect trout and salmon, whilst members of the subfamily *Orthohepevirinae*, further divided into four genera, *Paslahepevirus*, *Avihepevirus*, *Rocahepevirus* and *Chirohepevirus,* infect mammals and birds. The genus *Paslahepevirus* contains two species, *P. balayani* (previous species name: *Orthohepevirus A*), which include HEV variants able to infect humans and several mammalian species, and *P. alci*, which infects moose. Based on full-length genome analysis, eight distinct genotypes (gts) thus far have been proposed within the species *P. balayani* [7], with four major gts (1–4) implicated in human disease. Gt1 and gt2 infections are restricted to humans and cause large epidemics in developing countries due to poor sanitation and lack of clean drinking water, while gt3 and gt4 are zoonotic and cause sporadic and cluster cases of hepatitis E in both industrialized and developing countries [1,8]. Domestic pigs and wild boars represent the primary reservoirs of gt3 and gt4 HEVs [9], with consumption of raw or undercooked pork-meat products being considered the major cause of human infection [10]. Gt5 and gt6 have been identified only in wild boars in Japan [11], whilst gt7 and gt8 have been detected in dromedary camels in the United Arab Emirates [12] and in Bactrian camels in China [13], respectively. In addition to gt3 and gt4 HEVs, animal strains of genotypes 5, 7 and 8 from the species *P. balayani* [14,15] and rat HEVs from the species *Rocahepevirus ratti* (previous species name: *Orthohepevirus C*) are also known to have zoonotic potential [16,17,18,19,20] (Figure 1).

In recent years, the host range of *P. balayani* (*Orthohepevirus A*) has been significantly broadened by the identification of gt3 and gt4 strains from various animal species [22], including wild (Figure 2A) and domestic ruminants (Figure 2B). 

Deer have been addressed as potential reservoirs for HEV [23]. Indeed, the first identification of HEV RNA in wild ruminants also represents the first direct evidence of HEV zoonotic transmission, in which four human hepatitis E cases were epidemiologically linked directly to consumption of raw deer meat (Sika deer, *Cervus nippon nippon*). Molecular analysis of frozen leftover meat portions revealed the presence of HEV RNA, showing 100% nucleotide (nt) identity to the sequences identified in human patients [23]. Various surveys have subsequently explored HEV circulation in wild ruminants and provided the rationale for serological and molecular investigations in large and small domestic ruminants in order to understand their potential roles as HEV hosts, since livestock animals provide humans with essential sources of meat, milk and other dairy products, potentially constituting a source of exposure to HEV for humans. The aim of this review is to provide a general overview of the current knowledge on HEV epidemiology in ruminants, focusing on reported serological and molecular studies. Possible risk factors affecting HEV transmission among animals and their roles as potential sources of zoonotic infections are discussed.

## 2. Domestic Ruminants 

### 2.1. Bovids

The first evidence on the possible exposure of domestic bovids to HEV infection was documented in 1998 [24] in a multi-state serological investigation. Anti-HEV antibodies were detected in cows with prevalence rates of 29.0–62.0% in Somalia, Tajikistan and Turkmenistan, and of 12% in Ukraine [24]. Since then, several serosurveillance studies performed on cattle in Asia [25,26,27,28,29,30,31,32,33,34,35], America [36,37,38,39] and Africa [40,41,42,43,44,45] have revealed prevalence rates ranging from 1.4% to 47.0% (Table 1). Among domesticated wild bovids, seropositivity has also been reported in buffaloes in India [46] and in Egypt [40], with rates of 100.0% and 14.0%, respectively, and in 4.6% of bison (*Bison bison*) from the United States [38]. 

Although to lesser extents, different studies have also highlighted the presence of HEV RNA in bovids (Table 1). The first molecular genotyping in cattle was obtained in a molecular investigation performed in 2010 in China (Xinjiang Autonomous Region), in which HEV was detected in 8 out of 91 (8.8%) stool samples collected from seropositive dairy cows. On sequence analysis of a short region of the ORF2 gene, the HEV strains showed the highest nt identity (84.3–95.8%) to human and swine HEVs belonging to gt4 [50]. In a 2014 study, using specific molecular tools for gt4 HEV, viral RNA was found in 1.8% (3/167) of faecal samples collected from domesticated yaks (*Bos grunniens*) reared in Tibet Region for meat and milk [58]. The complete sequence analysis of one such strain revealed the highest nt identity to swine (99.1%) and human (93.8%) strains previously detected in China. Furthermore, HEV-positive yaks (3/92, 3.3%) were found only in Qinghai, one of the two provinces (Gansu and Qinghai) investigated. Accordingly, the authors claimed that a neighboring pig farm was the potential source of HEV transmission to yaks [58]. Subsequent molecular studies reported the detection of HEV RNA, with rates of 3.0% (8/254) in serum samples collected from yellow cattle (*Bos taurus*) of local breeds in the Shandong Province of Eastern China and of 37.1% (52/140) in stools of Holstein cows in Yunnan Province (Southwest China) [33,52]. In a 644 bp fragment of the ORF2 gene, gt4 HEV sequences detected in yellow cattle and in Holstein cows shared 83.3–85.3% nt identity to each other, clustering within two different subtypes, d and h [33,52]. The presence of viral RNA was also demonstrated in cow milk samples. The infectivity of contaminated raw milk and even of pasteurized milk was confirmed by experimental infection of rhesus macaques. Go et al. (2019) reported the first identification of HEV in bovine liver (1.0%, 1/100) purchased from local grocery markets between February 2017 and July 2018 in Seoul (Korea) [56]. 

Despite the several studies performed in European countries [49,51,53,54], to date, information on the possible circulation of HEV in cattle has been documented only in one survey conducted in Turkey [57]. In this study, viral RNA was found in raw cow milk, with a prevalence rate of 29.2%. To explain the discrepancies observed between the European (Figure 3A) and Asiatic (Figure 3B) studies [33,52], it was speculated [53,54] that in the Asiatic countries small-sized farms with mixed animals could generate a higher risk for HEV transmission from pigs to cattle. More recently, in a molecular survey performed in Egypt on a non-mixed dairy farm, a gt3 HEV subtype a strain was identified in a cow milk sample, with confirmed seroconversion of some animals at the follow-up [44].

### 2.2. Goats

The susceptibility of goats to HEV infection has been assessed in several serological and molecular surveys (Table 2). 

The first evidence was collected in a study from Turkmenistan [24], in which HEV antibodies were detected in 67.0% of the goats tested. Subsequent investigations conducted worldwide [27,30,31,40,41,43,46,47,59,61,62,63,64,66,67] reported antibody-detection rates ranging from 0.6% to 100.0%. Interestingly, in a study conducted in Virginia, USA, IgG anti-HEV antibodies were found in 16.0% (13/80) of the goats tested, also demonstrating the presence of neutralizing antibodies to HEV in selected IgG anti-HEV positive goat sera [59]. However, attempts to infect goats experimentally with three different HEV genotypes (gt1-Sar-55, gt3-Meng and gt4-TW6196E) were unsuccessful [59]. Direct evidence on the circulation of HEV in goats was documented for the first time in a European study [60], in a survey of goat farms located in a restricted geographical area of Italy (Abruzzo, Southern Italy). HEV RNA was found in 11 out of 119 goat faecal samples, with an overall prevalence of 9.2%. On sequence analysis of an 800 nt-long fragment of the ORF2 gene, four strains were closely related to animal and human gt3 HEVs, subtype c, with the highest identity (94.2–99.4% nt) to a wild boar strain identified in the same geographical area and where the density of free-ranging wild boar populations was high [60]. Shortly after, in 2017, HEV RNA was identified in stool, serum and milk samples collected from goats in Yunnan Province in China [61]. HEV prevalence in the goat faecal samples ranged from 60.0% to 74.0%—values significantly higher than that found for cows (37.1%) in the same geographical area [52]. A total of 15/28 (53.6%) goats were found to be viremic. In addition, HEV RNA was detected in all milk samples collected from each HEV-infected goats, with viral loads comparable to those found in the faeces or sera. Immunohistochemistry revealed the presence of HEV antigens in the livers and spleens of infected goats. Sequence analysis of goat strains showed a high genetic relatedness (>99.6%) to human and animal HEVs classified within genotype 4 subtype h and previously detected in the same geographic area [61]. Gt4 strains were also identified in liver samples from slaughtered goats (4.0%, 2/50) in a study performed in the Tai’an region of China [62]. On analysis of the partial HEV ORF2 sequences, the two strains showed the highest nt identity (92.5–93.0%) to bovine strains detected in 2016 in the same province (data unpublished). The presence of HEV in goat milk samples was also reported in Egypt, with an overall prevalence of 0.7% [63]. The identified strains, characterized as gt3 HEV subtype a, showed a high nt identity to strains of bovine and human origin previously detected in the same geographical area [44]. Furthermore, in the same survey, HEV RNA was also detected in fresh liver samples of two seropositive goats [63]. The presence of HEV RNA in goat milk samples has been further documented in two European studies conducted in Turkey [57] and in the Czech Republic [65], with prevalences of 18.5% and 1.4%, respectively.

### 2.3. Sheep

Initial evidence on the susceptibility of sheep to HEV infection was reported in 1994, in an experimental study which aimed to establish HEV animal models. The inoculated lambs developed clinical, biochemical and histological findings consistent with hepatitis, showing virus shedding in faeces and the presence of HEV RNA in the parenchymal organs [68]. In a 1998 study [24], anti-HEV antibodies were detected in 42.0% of sheep from Turkmenistan, demonstrating that this animal species was naturally exposed to HEV or to a related virus. In a 2007 Indian study [46], 58 sheep sera collected from abattoirs in Lucknow (India) tested for anti-HEV IgG using two different enzyme-linked immunoassays (EIAs) showed 100.0% and 77.5% positivity, respectively. However, an inhibition assay for the stool suspension of a patient positive for HEV RNA was not able to confirm the anti-HEV specificity of the antibodies detected in the sheep [46]. Subsequent independent serological investigations (Table 3) reported anti-HEV IgG in sheep, with rates of 9.3–35.2% in China [28,29,31,32,69,70], 1.9–2.1% in Spain [47,66], 4.4% in Egypt [40], 10.5–31.8% in Nigeria [41,71], about 21.0% in Italy [64,72], 12.0% in Burkina Faso [43], 16.6% in Portugal [73] and 12.7% in Jordan [35]. Genetic information on the HEV strains circulating in sheep (Table 3) was first obtained in the Xinjiang region of China [74]. On sequence analyses of a short region of the ORF2 gene of six strains (11.1%, 6/54) detected in faecal samples, the highest genetic identities (84.67–95.36%) were found to bovine, swine and human HEVs belonging to gt4 [74]. HEV RNA was also detected in 4 out of 75 (5.3%) livers collected from slaughtered sheep [70]. Circulation of HEV gt4 in sheep in China was further confirmed in a 2016 molecular study [33] conducted in a rural area of Shandong Province (Eastern China). On sequence analysis of a 644 nt fragment of the ORF2 gene obtained from 8 out of 70 (11.4%) serum samples of domestic sheep, high nt identity (95.1–99.8%) was found to HEV sequences detected in yellow cattle tested in the same survey from mixed farms [33]. In Europe, in a molecular study performed in Southern Italy [72], gt3 HEV RNA was detected in 20/192 (10.4%) sheep faecal samples. Interestingly, 3/20 HEV-positive animals were also viremic. On sequence analysis, the sheep HEV strains were found to be similar to HEV strains previously identified in goats, wild boars and human patients in the same region (Abruzzo) [72]. Circulation of HEV among sheep populations in Italy has been also reported in another geographical area (Northwestern Italy). HEV RNA was detected by qRT-PCR in 4/134 (3.0%) faecal specimens collected from two sheep herds serologically positive for HEV [64]. As observed for cows and goats [44,52,57,61,63], sheep milk could represent a potential source of HEV infection for consumers, since HEV RNA has been detected in Turkey [57] and in the Czech Republic [65], in 12.3% and 1.4%, respectively, of sheep milk samples analyzed. 

## 3. Wild Ruminants 

The discovery of HEV in wild ruminants dates back to 2003 in Japan [23], during an outbreak of acute hepatitis affecting four members of the same family, all of whom had consumed raw deer meat (sika deer, *Cervus nippon*). All patients serotested positive for HEV RNA, IgM and anti-HEV IgG. Furthermore, molecular analysis of frozen leftover meat portions revealed the presence of viral RNA showing 100% nt identity to the sequences identified in the human patients [23]. Since then, several serological and molecular investigations have been conducted in different species of wild ruminants belonging to the *Cervidae* and *Bovidae* families, demonstrating their important roles as HEV hosts. Studies mainly focused on cervid species, commonly referred to as deer, have revealed seroprevalence rates ranging from 0.2 to 12% in Asia [75,76,77,78,79,80,81], 1.7% to 62.7% in America [82,83,84] and 0.4% to 19.5% in Europe [64,85,86,87,88,89,90,91,92,93,94,95,96,97,98,99,100,101], with reported HEV RNA prevalences of 0.06–35.0% [78,79,102], 0% [84] and 1.2–34.1% [48,49,51,85,86,88,89,91,92,93,94,95,96,97,98,103,104,105,106,107,108,109,110], respectively. Besides deer, anti-HEV antibodies have been detected in 5.9% of muskoxens (*Ovibos moschatus*) in Norway [100] and in 1.2–5.1% of chamois (*Rupricapra rupricapra*) [98,101], 6.3% of Alpine ibex (*Capra ibex*) [64] and 1.2% of mouflons (*Ovis aries musimon*) [101] in Italy. In a Czech Republic study, 5 out of 39 (12.8%) faecal samples of mouflons hunted or living in game enclosures tested positive for HEV RNA [104]. In Iran, a molecular prevalence of 6% was found in Persian gazelle (*Gazella subgutturosa*) [111] (Table 4).

To date, HEV strains identified in wild ruminants belong to gt3 [48,85,88] subtypes 3a [49], 3b [78], 3e [49,95,109,110], 3f [94] and 3i [107] and to gt4 [79,102]. 

HEV RNA has been searched for in different biological materials, including various organs (e.g., liver, spleen, kidney and muscle), serum, faeces and bile. The presence of viral RNA has been repeatedly confirmed in liver, with detection rates between 1.7% and 80.0% [48,49,51,78,86,91,93,94,105,107,110]; in serum, with rates ranging from 0.06% to 60.0% [78,84,87,90,92,94,106]; and in feces, with rates of 1.2–50.0% [102,104,107]. In an Italian study [110], HEV RNA was identified in liver specimens collected from roe deer (*Capreolus capreolus*) (10.4%; 5/48) and from a fallow deer (*Dama dama*) (1.7%; 1/60). HEV antigens were detected in the fallow deer by immunohistochemistry and associated with degenerative and inflammatory lesions with predominantly CD3+ cellular infiltrates and hyperplasia of Kupffer cells. Rutjes et al. (2010), in the Neatherlands, identified qRT-PCR HEV RNA in the muscle (5.0%) of red deer (*Cervus elaphus*) [86]. In a study conducted in Germany [93], HEV RNA was detected in the livers (50.0%), sera (60.0%), muscles (100.0%), spleens (50.0%) and kidneys (50.0%) of two red deer and five roe deer. Taken together, the detection of HEV RNA in deer liver and in other organs, especially in muscle tissue, highlights the risk of transmission of HEV to humans through consumption of undercooked meat of these animals. However, in general, lower serological and molecular prevalences have been found in wild ruminants than in wild boars living in the same territories [78,93]. Moreover, using quantitative assays, the viral loads found in serum, faeces and organ samples seem to be lower in deer than in wild boars and pigs [9,93,104]. In the liver, viral load can range between 20 and 10^7^ RNA copies/g in pig and, similarly, between 40 and 10^8^ RNA copies/g in wild boar [9]. In contrast, HEV load in deer liver seems markedly lower, ranging between 12 and 2000 RNA copies/g [93].

In a 2014 study in Sweden, a novel HEV-like virus was discovered in the liver sample of a dead moose. The obtained sequence of approximately 5100 nt was found to be highly divergent genetically, sharing only 46.1–63.1% nt identity with other HEV strains sequenced thus far [103]. This virus has been classified in the species *Paslahepevirus alci* of the genus *Paslahepevirus* [6]. In subsequent investigations conducted in Sweden, the RNA of a HEV-like moose virus was identified in 15.0% (34/231) and 18.2% (12/66) of moose liver samples, but not in humans or wild boars [89,92]. The zoonotic potential of the moose HEV-like virus remains unclear.

## 4. Discussion

The detection of anti-HEV antibodies and the presence of RNA in serum rather than in faecal and/or in liver samples in ruminant species are clues that these animals are susceptible to HEV infection. Detection of negative-stranded RNA replication intermediates in infected tissues or experimental infection studies could provide additional pieces of evidence to help understand whether ruminants are true hosts for HEV infection or just suboptimal hosts [22]. The experimental infection of lambs with a human HEV strain (of unknown genotype) provided clinical evidence consistent with acute hepatitis [68]. However, biochemical and/or histological alterations have only been observed in HEV-infected goats [61] and in the liver of a fallow deer with natural HEV infection detected by quantitative RT-PCR [110]. In several studies, only anti-HEV antibodies were detected, whilst the source of HEV infection was not determined, suggesting merely exposure to HEV or a HEV-related agent [22]. In a prospective study conducted in a seropositive dairy herd, monitoring of newborn calves from birth to six months of age revealed seroconversion to IgG anti-HEV. However, despite several attempts, using either broad-spectrum RT-PCR assays or a next-generation sequencing approach, viral RNA was not detected. It was hypothesized that cattle may be susceptible to antigenically related strains, still genetically uncharacterized, inducing cross-reactive HEV antibodies [39]. A similar finding had been previously described for goats in the study of Sanford et al. [59], in which seroconversion was observed in 7 out of 11 kids monitored from birth until 14 weeks of age, although HEV RNA was not detected in faecal or serum samples [59]. Hence, interpreting HEV serological data could be limited just by the inability of the molecular tools that are currently available to detect genetically highly divergent HEV strains, as observed in the study of Geng et al. [31], where none of the antigen-positive goat or cattle samples tested positive for HEV RNA. In addition, as previously discussed by Yugo et al. [39], limits on viral molecular detection from ruminant samples due to the presence of amplification inhibitors should be considered, especially during sample processing or nucleic-acid extraction.

Frequent detection with a high prevalence of specific HEV genotypes in the same species in different geographical areas clearly indicates a true animal reservoir, as exemplified by pigs and wild boars [9,22]. By contrast, in domestic and wild ruminants, HEV has been identified sparsely in most of the cases, with low serological and molecular prevalence rates, suggesting that these animal species are not true reservoirs of HEV but, more likely, that they may be infected occasionally due to spillover events. This hypothesis is also supported by the recurrent detection in cattle, sheep, goats and several wild ruminants of gt3 and gt4 HEV strains genetically highly related to HEVs identified in pigs or wild boars in the same geographical areas [33,50,58,60,61,74]. In domestic ruminants, HEV prevalence appears to be higher in rural areas with traditional mixed farming systems consisting of family-based small-sized farms hosting pigs and other domestic animals—a favorable epidemiological picture that may foster inter-species interactions [26,31,49,58,60]. In a recent study in Burkina Faso [45], cattle seropositivity was significantly associated with the presence of pigs on the same farm, suggesting the role of swine as risk factors [45]. However, this is not always the case, since Geng et al. [55] did not detect HEV in the faeces and milk of cows reared on mixed farms or neighboring farms with pigs in the Hebei province of China [55], nor in Belgium, where viral RNA was not detected in cow fecal and milk samples despite being collected on a mixed farm in which HEV infection of swine had been demonstrated [54]. In addition, Gt3 HEV strains were detected in cows and goats of Assiut Village, Egypt, where pig farms are not common; even in the area housing pig farms, mixing of pigs and cows is uncommon due to religious beliefs [44,63]. Accordingly, HEV transmission to domestic ruminants could be due to several factors, including husbandry practices, type and intensity of inter-species contacts as well as hygiene on the farms and handling/management of HEV-contaminated manure produced by different animal species [44,45]. In a survey performed in the Czech Republic [65], it was speculated that pastures contaminated by HEV-positive wild animals (wild boars, red deer, roe deer and mouflons) could be the source of infection for small ruminants in which HEV RNA was detected in raw milk samples [65]. In a study conducted in Jordan [35], several farm-management practices were significantly associated with HEV seroprevalence at the farm level. Large and small dairy ruminant farms that reported infrequent cleaning of feeder stations and infrequent general farm cleaning or mixing small ruminants (sheep and goats) together in the same flock had greater odds of HEV seroprevalence. 

For wild ruminants, the supplementary winter-feeding in rural areas might play a predominant role in the spread of HEV. Inter-species grouping between wild boars and wild ruminants across the supplementary feeding sites could favor HEV transmission in deer through the consumption of feed contaminated by the faeces of wild boar [49]. Interestingly, surveys conducted on farmed deer, in areas to which wild boars had no access, revealed a low HEV seroprevalence, indicating that HEV infection in deer could be modulated by contact with HEV-infected wild boar [85,96,112]. In the study of Takahashi et al. (2004) [113], full-length genome sequencing of HEV strains from wild boar and deer showed a nt identity of 99.7%, suggesting that inter-species transmission of HEV may also occur in nature without the help of feeding sites [113]. Italian studies in a restricted area of the Tuscan–Emilian Apennines identified HEV antibodies in 13.9% (35/251) of the analyzed deer sera [95] and HEV RNA in 25% of wild boar bile samples [114]. In a study conducted by Tomiyama et al. (2009) [80], in two pig farms from Japan, Hidaka District, with free-range grazing animals, HEV-positive samples were obtained from deer, suggesting that deer were infected through ingestion of pasture contaminated with the faeces of the HEV-infected free-ranging pigs [80]. However, in another study [98], anti-HEV antibodies were found in a red deer from the Stelvio National Park (Italy), where wild boars are not present, and in a chamois—an animal living at high altitudes and with minimal interactions with other animal species. In these cases, direct or indirect contact with wild boar is unlikely to account for exposure to HEV. Even though a lack of specificity of the serological assay employed could be considered, as was also observed in the study of Rutjies et al. [86], in which an overestimated prevalence for deer species was suspected, it cannot be excluded that there could be unknown sources of HEV infection for wild ruminants [98]. Rinaldo et al. [99] suggested the possibility that hares and rats could potentially have been involved in the transmission of HEV to Norwegian semi-domesticated reindeer in an area in which wild boars have never been observed [99]. The roles of rodents in HEV infections have been repeatedly investigated, mostly on pig farms due to their potential abundance on farms. In a review analyzing infection dynamics and persistence of hepatitis E virus on pig farms, it was discussed that the low prevalence of HEV gt3 in rodents around farms and the detection predominantly in intestines support the argument that rodents are only accidental hosts of HEV gt3. With probability, rodents may be considered potential risks that mechanically contribute to the spread of infected faecal material and thereby to environmental contamination [115]. Overall, both for domestic and for wild ruminants, additional sources of exposure to HEV could include contaminated water and environmental contaminations due to faecal shedding by infected animals or due to the use of swine manure in agriculture [64]. 

Evidence for HEV zoonotic transmission by ingestion of uncooked swine, wild boar or deer meat products have been well documented [23,116,117,118,119,120]. Among ruminants, in addition to cluster cases directly linked to the consumption of raw deer meat, there is growing evidence on the role of domestic camels as zoonotic sources of HEV gt7 infection [121], as indicated by a case of chronic hepatitis E after transplantation in a patient from the United Arab Emirates who regularly consumed camel milk and meat [20]. Human HEV infections associated with consumption of products from domestic cattle, goats and sheep have not been reported so far. However, detection of HEV RNA in liver [56,62,70] and milk [44,52,57,61,63,65] samples collected from cows, goats and sheep make this possibility tangible. In addition, the close phylogenetic relationships observed between human and ruminant HEV strains give some support to this hypothesis [44,52,60,61,72]. As Sayed and El-Mokhtar [122] previously discussed, ingestion of ruminant raw milk products and undercooked liver should be considered potential risk factors for HEV transmission to humans. Screening to identify the risk of HEV infection through the consumption of these ruminant products should be routinely performed, especially for products sold in local markets, but also for other edible ruminant organs, including intestines, kidneys, brains and spleens [122]. Furthermore, especially in rural communities, where households typically own a small number of cows, sheep and/or goats raised for milk and meat production intended for personal consumption, there could be a high exposure to animal contact [122]. Indeed, these species, small ruminants in particular, are social and docile, and these attributes may favor close contact and interactions, increasing the risks of zoonotic infection. The risk of HEV transmission to Egyptian goat owners was assessed and none of the households owning seronegative goats had HEV markers, whilst 80% of households owning seropositive goats also tested positive for anti-HEV IgG, suggesting a potential interaction between goats and households. In detail, in this study, a possible goat-to-human transmission was considered [63], whilst a possible transmission from humans to goats, although it could not be completely excluded, was not discussed. However, in a survey conducted in Lao village [34], higher anti-HEV IgG seroprevalence has been reported in cattle farmers compared to other villagers [34], thus supporting the possible animal-to-human transmission route. The potential occupational risks of HEV infection in settings with exposure to domestic ruminants have been assessed only in two studies so far. In the serosurvey performed by Wu et al. [69], a seroprevalence of 57.7% (15/26) was detected in butchers working in the same slaughterhouse in which sheep tested positive for the presence of anti-HEV antibodies and for the presence of viral RNA in liver [69]. More recently, the occupational risk was investigated in shepherds and sheep-milk-cheesemakers in a study in Portugal [73], which revealed that the seroprevalence in workers was significantly higher (29.3%) when compared with the population controls (16.1%). Similarly, several serological studies have suggested that populations of workers having occupational contact with wild animals, mainly represented by wild boars and deer, have a higher seroprevalence than the related general populations [123,124,125,126,127,128,129]. The exposure risk to HEV infection in forestry workers was documented in studies performed in Germany and France in which higher HEV seroprevalence rates were found in these groups than in the control groups [123,124,125]. 

## 5. Conclusions

In conclusion, whether domestic and wild ruminants are involved in spillover or are true reservoirs of HEV needs to be further investigated. Animal experiments are required to address this hypothesis. Detailed and large observational studies worldwide, based on standardized and improved diagnostic methods specific for HEV, would be useful to understand better the role of ruminants in HEV epidemiology and the risk of transmission to humans. A valuable approach could be to evaluate the presence of HEV by combining serological and molecular investigations in animal populations, possibly screening different biological fluids and tissues, including sera, stools, milk and tissue samples from livers, muscles, and spleens. Furthermore, as several studies have demonstrated the association between direct contact with swine and higher HEV seroprevalences in professionally exposed persons [130,131,132], additional serological surveillance for HEV among individuals with occupational exposure to ruminants, such as slaughter-plant workers, veterinarians, farmers and hunters, would be helpful in better understanding the role of ruminants as HEV hosts and the occupational risks linked to contact with them. In addition, surveillance plans to ascertain the viral hazards for humans associated with the consumption of ruminant-derived products should be devised and enacted. For this purpose, current methods of nucleic-acid extraction and purification, especially for food matrices, should be improved in order to avoid the decreased sensitivity of the molecular techniques employed. Finally, unbiased analysis of nucleic acids from different matrices of animal origin may constitute an additional tool to increase the frequency of HEV detection and identify still-unknown HEV-like viruses.

## Figures and Tables

**Figure 1 pathogens-11-01124-f001:**
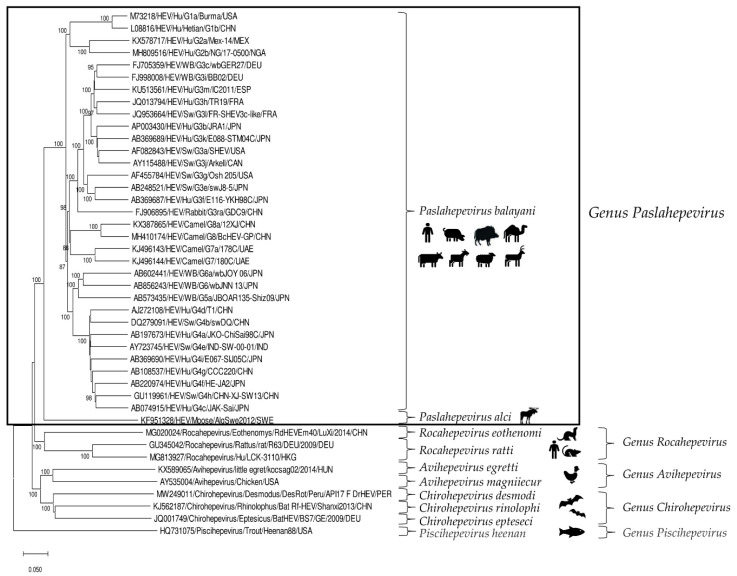
Phylogenetic tree constructed on the basis of the complete nucleotide sequences of the reference strains representative of the family *Hepeviridae* (http://talk.ictvonline.org accessed on 23 September 2022). The tree was generated using the neighbor-joining method and the p-distance model supplying a statistical support with bootstrapping of 1000 replicates. The scale bar indicates nucleotide substitutions per site. Evolutionary analyses were conducted in MEGA X [21]. Hu: human; SW: swine; WB: wild boar.

**Figure 2 pathogens-11-01124-f002:**
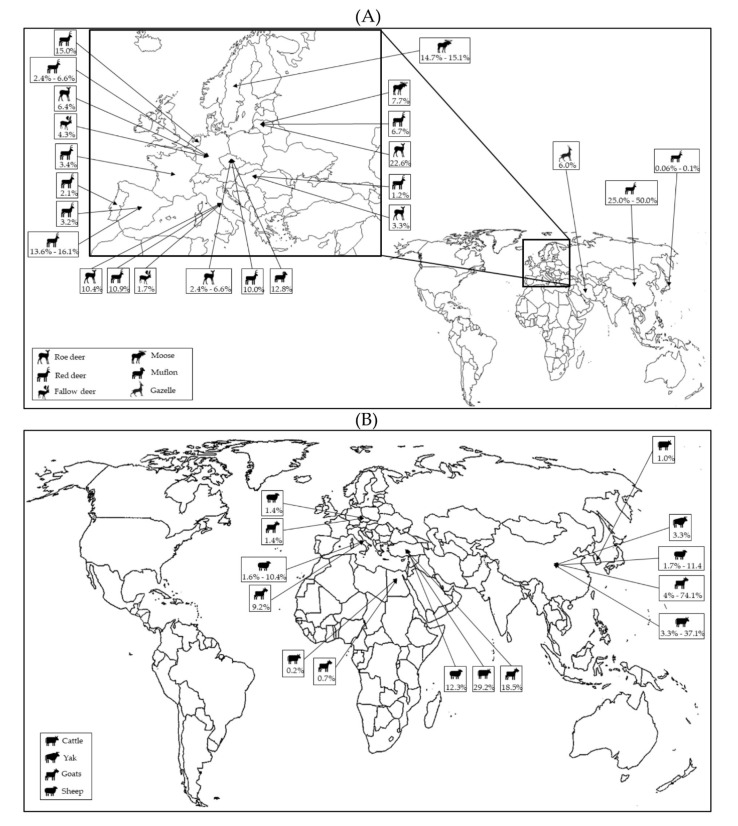
Global distribution of HEV molecular studies performed in wild (**A**) and domestic (**B**) ruminants.

**Figure 3 pathogens-11-01124-f003:**
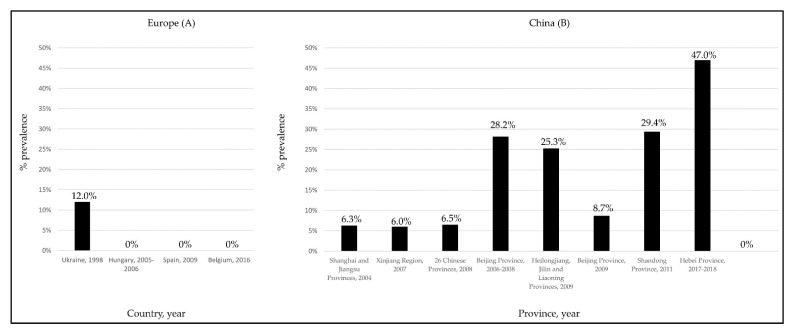
HEV seroprevalence rates detected in cattle in European (**A**) and Chinese (**B**) surveys.

**Table 1 pathogens-11-01124-t001:** Serological and molecular prevalence of HEV in domestic bovids.

Animal Species	Country	Year	Type of Sample	Seroprevalence % (Positive/Total)	Molecular Prevalence % (Positive/Total)	Genotype (gt)	Reference
*Bos taurus*	Ukraine	1998	Serum	12.0%	^1^ n.d.		[24]
	India	1994 1999	Serum	4.0% (4/91) 6.9% (13/188)	n.d.		[25]
	China	2002	Serum	6.3% (12/190)	n.d.		[26]
	Brazil	2005	Serum	1.4% (1/70)	n.d.		[37]
	China	2004	Serum	6.0% (6/100)	0% (0/100)		[27]
	China	2009	Serum	18.7% (301/1612)	3.3% (3/120)		[28]
	China	2009	Serum	29.4% (54/184)	n.d.		[29]
	Spain	2009	Serum	0% (0/1170)	n.d.		[47]
	Hungary	2005–2006	Faeces liver	n.d.	0% (0/30) 0% (0/2)		[48]
	Hungary	2005–2009	Faeces	n.d.	0% (0/125)		[49]
	China	2007	Serum	6.5% (13/200)	0% (0/200)		[30]
	China	2008	Serum	28.2% (257/912)	0% (0/912)		[31]
	China	2010	Faeces	n.d.	8.8% (8/91)	gt4	[50]
	China	2006	Serum Milk	25.3% (66/261) 14.9% (40/269)	0% (0/261) 0% (0269)		[32]
	USA	2011	Serum	15.0% (174/1156)	0% (0/174)		[38]
	Egypt	2010	Serum	21.6% (11/51)	n.d.		[40]
	Croatia	2009	Blood, spleen, liver	n.d.	0% (0/30)		[51]
	China	2011	Serum	47.0% (120/254)	3.0% (8/254)	gt4	[33]
	Nigeria	2014	Serum	0% (0/37)	n.d.		[41]
	China	2015	Faeces, serum, milk	n.d.	Faeces: 37.1% (52/140) Milk: 100.0% (52/52) *	gt4	[52]
	Germany	2008	Bulk milk	n.d.	0% (0/400)		[53]
	Nigeria	2018	Serum	0% (0/30)	n.d.		[42]
	Vietnam	2015	Faeces	6.8% (11/161)	0% (0/120)		[34]
	Belgium	2016	Milk	0% (0/275)	0% (0/1559)		[54]
	USA	2009	Faeces	20.4% (200/983)	0%		[39]
	Burkina Faso	2019	Serum	26.4% (19/72)	n.d.		[43]
	China	2017	Faeces, retail milk	0% (0/416)	0% (0/883)		[55]
	Korea	2017–2018	Liver	n.d.	1.0% (1/100)	gt4	[56]
	Turkey	2016	Milk	n.d.	29.2% (14/48)	gt1, gt3, gt4	[57]
	Jordan	2020	Serum	14.5% (18/124)	n.d.		[35]
	Egypt	2017	Faeces, milk	1.7% (8/480)	0.2% (1/480)	gt3	[44]
	Burkina Faso	2017	Serum	5.1% (24/475)	n.d.		[45]
*Bos grunniens* (yak)	China	2013	Faeces	n.d.	1.8% (3/167); Qinghai Province 3.3% (3/92), Gansu Province 0% (0/75)	gt4	[58]
Buffaloes	India	2007	Serum	100.0% (30/30)	n.d		[46]
*Bison bison*	USA	2011	Serum	4.6% (3/65)	0% (0/3)		[38]
*Syncerus caffer*	Egypt	2010	Serum	14.0% (8/57)	n.d.		[40]

^1^ n.d.: not determined; *: cohort of selected animals.

**Table 2 pathogens-11-01124-t002:** Serological and molecular prevalence of HEV in goats.

Country	Year	Type of Sample	Seroprevalence % (Positive/Total)	Molecular Prevalence % (Positive/Total)	Genotype (gt)	Reference
Turkmenistan	1998	Serum	67.0%	^1^ n.d.		[24]
India	1994 1999	Serum Serum	0% (0/52) 0% (0/188)	n.d. n.d.		[25]
China	2002	Serum	0% (0/316)	n.d.		[26]
Brazil	2005	Serum	0% (0/5)	n.d.		[37]
India	2007	Serum	100.0% (98/98)	n.d		[46]
China	2004	Serum	24.0% (12/50)	0% (0/50)		[27]
Spain	2009	Serum	0.6% (7/1143)	n.d.		[47]
China	2007	Serum	7.5% (15/200)	0% (0/200)		[30]
China	2008	Serum	10.4% (73/700)	0% (0/700)		[31]
USA	2002	Serum, faeces	16.0% (13/80)	0% (0/80)		[59]
Egypt	2010	Serum	9.4% (3/32)	n.d.		[40]
Nigeria	2014	Serum	37.2% (16/43)	n.d.		[41]
Italy	2012	Faeces	n.d.	9.2% (11/119)	gt3	[60]
China	2015	Faeces, serum, milk	IgM: 3.6% (1/28) IgG: 14.3% (4/28)	2015 faeces: 74.1% (40/54) 2016 faeces: 60.0% (12/ 20) Serum: 53.6% (15/28) Milk: 100.0% (4/4) *	gt4	[61]
China	2017	Serum, liver	46.7% (50/120)	4.0% (2/50)	gt4	[62]
Burkina Faso	2015	Serum	28.4% (23/81)	n.d.		[43]
Nigeria	2018	Serum	0% (0/26)	n.d.		[39]
Turkey	2016	Milk	n.d.	18.5% (12/65)	gt1, gt3, gt4	[57]
Egypt	2017	Milk	7.1% (20/280)	0.7% (2/280)	gt3	[63]
Italy	2017	Serum, faeces	11.4% (19/167)	0.0% (0/167)		[64]
Czech Republic	2019	Milk	n.d.	1.4% (4/290)		[65]
Spain	2015	Serum	13.8% (33/240)	0% (0/240)		[66]
China	2014	Serum	26.1% (47/180)	n.d.		[67]

^1^ n.d.: not determined; *: cohort of selected animals.

**Table 3 pathogens-11-01124-t003:** Serological and molecular prevalence of HEV in sheep.

Country	Year	Type of Sample	Seroprevalence % (Positive/Total)	Molecular Prevalence % (Positive/Total)	Genotype (gt)	Reference
Turkmenistan	1998	Serum	42.0%	^1^ n.d.		[24]
Brazil	2005	Serum	0% (0/12)	n.d.		[37]
India	2007	Serum	100.0% (58/58) 78.0% (47/58)	n.d		[46]
China	2004–2006	Serum	9.8% (20/207)	n.d		[29]
Spain	2009	Serum	1.9% (36/1357)	n.d.		[47]
China	2009	Serum	12.4% (162/1302)	1.7% (2/115)		[28]
China	2010	Serum	9.9% (33/334)	n.d.		[31]
China	2010	Serum	28.9% (142/490)	n.d.		[69]
China	2010	Faeces	n.d.	11.1% (6/54)	gt4	[74]
China	2006	Serum	9.3% (53/541)	0% (0/541)		[32]
Egypt	2010	Serum	4.4% (2/45)	n.d.		[40]
Nigeria	2012	Serum	10.5% (2/19)	n.d.		[41]
China	2014	Serum, liver	35.2 % (176/500)	5.3% (4/75)	gt4	[70]
China	2011	Serum	32.0% (70/222)	11.4% (8/70)	gt4	[33]
Nigeria	2017	Serum	31.8% (56/176)	n.d.		[71]
Italy	2018	Serum, faeces	21.3% (40/192)	Faeces: 10.4% (20/192) Serum: 1.6% (3/192)	gt3	[72]
Burkina Faso	2019	Serum	12.0% (9/75)	n.d.		[43]
Turkey	2016	Milk	n.d.	12.3% (8/65)	gt1, gt3, gt4	[57]
Italy	2017	Serum, faeces	21.6 % (29/134 )	3.0% (4/134)		[64]
Jordan	2020	Serum	12.7% (26/205)	n.d.		[35]
Czech Republic	2019	Milk	n.d.	1.4% (4/290)		[65]
Portugal	2016	Serum	16.6% (15/90)	n.d		[73]
Spain	2015–2017	Serum	2.1% (5/240)	0% (0/240)		[66]

^1^ n.d.: not determined.

**Table 4 pathogens-11-01124-t004:** Serological and molecular prevalence of HEV in wild ruminants.

Animal Species	Scientific Name	Country	Type of Sample	Seroprevalence % (Positive/Total)	Molecular Prevalence % (Positive/Total)	Genotype (gt)	Reference
Sika deer	*Cervus nippon*	Japan	Serum, liver	2.0% (2/117)	0% (0/117)		[75]
		Japan	Serum, faeces, liver	2.6% (25/976)	0% (0/501)		[76]
		USA	Serum	0% (0/174)	n.d.		[82]
		China	faeces	^1^ n.d.	25.0% (2/8)	gt4	[102]
		Poland	Serum	0% (0/68)	n.d.		[87]
		China	Serum	5.4% (46/847)	n.d.		[77]
		Germany	Serum, liver	0% (0/12)	0% (0/12)		[96]
		Japan	Serum, liver	4.3% (17/395)	0.2% (1/199)	gt3	[78]
		Japan	Serum	0.04% (1/2250)	0.06% (1/1688)	gt4	[79]
Yezo deer	*Cervus nippon yesoensis*	Japan	Serum	34.8% (181/250)	n.d.		[80]
Roe deer	*Capreolus capreolus*	Hungary	Faeces, liver	n.d.	34.1% (11/32)	gt3	[48]
		Hungary	Faeces, liver	n.d.	22.0% (9/41)	gt3	[49]
		Netherlands	Serum, faeces, Liver, muscle	0% (0/8)	0% (0/8)		[86]
		Poland	Serum	0% (0/38)	n.d.		[87]
		Belgium	Serum, liver	3%	0% (0/27)		[94]
		Italy	Liver	n.d.	0% (0/30)		[106]
		Czech Republic	Faeces	n.d.	3.3% (1/30)		[104]
		Croatia	Blood, spleen, liver	n.d.	0% (0/40)		[51]
		Germany	Serum, liver	2000–2001: 6.8% 2012–2013: 5.4%	2000–2001: 0% 2012–2013: 0%		[91]
		Sweden	Serum, faeces	6.9% (2/29)	0% (0/29)		[92]
		Germany	Serum, liver, muscle, spleen, kidney	0% (0/59)	6.4% (5/78)	gt3	[93]
		Lithuania	Serum, liver	n.d.	22.6% (21/93)	gt3	[107]
		Italy	Serum, faeces	3.1% (1/32)	0% (0/32)		[64]
		Italy	Liver	n.d.	0% (0/6)		[108]
		Norway	Serum	0% (0/86)	n.d.		[100]
		Italy	Serum	0.4% (1/227)	n.d.		[101]
		Italy	Liver	n.d.	10.4% (5/48)	gt3	[110]
Red deer	*Cervus elaphus*	Spain	Serum	10.4% (101/968)	13.6% (11/81)	gt3	[85]
		Hungary	Faeces, liver	n.d.	10.0% (3/30)	gt3	[49]
		Netherlands	Serum, faeces, Liver, muscle	5.0% (2/38)	15.0% (6/39)		[86]
		Poland	Serum	0% (0/118)	n.d.		[87]
		Italy	Serum	5.6% (3/54)	n.d.		[90]
		Belgium	Serum, liver	1%	3.4% (1/29)	gt3	[94]
		Spain	Serum	12.9% (9/70)	16.1% (13/81)	gt3	[88]
		Italy	Serum	13.9% (35/251)	10.9% (10/91)	gt3	[95]
		Czech Republic	Faeces	n.d.	1.2% (2/169)		[104]
		France	Liver, bile	n.d.	3.2% (2/62)		[105]
		Croatia	Blood, spleen, liver	n.d.	0% (0/280)		[51]
		Germany	Serum, liver	2000–2001: 2% 2012–2013: 3.3%	2000–2001: 1.9% 2012–2013: 6.6%		[91]
		Sweden	Serum, faeces	7.1% (1/14)	0% (0/14)		[92]
		Germany	Serum, liver, muscle, spleen, kidney	0% (0/78)	2.4% (2/83)	gt3	[93]
		Lithuania	Serum, liver	n.d.	6.7% (1/15)		[107]
		Germany	Serum, liver	0% (0/23)	0% (0/22)		[96]
		Italy	Serum	0.8% (2/255)	0% (0/255)		[98]
		Italy	Serum, faeces	2.6% (1/38)	0% (0/38)		[64]
		Finland	Serum	0% (0/12)	0% (0/12)		[97]
		Italy	Liver	n.d.	0% (0/218)		[108]
		Norway	Serum	4% (7/177)	n.d.		[100]
		Portugal	Faeces	n.d.	2.1% (2/95)	gt3	[109]
		Italy	Serum	0% (0/96)	n.d.		[101]
Fallow deer	*Dama dama*	Poland	Serum	0% (0/5)	n.d.		[87]
		Germany	Serum	0% (0/46)	4.3% (2/46)		[91]
		Germany	Serum	0% (0/22)	0% (0/22)		[93]
		Germany	Serum, liver	0% (0/73)	0% (0/72)		[96]
		Portugal	Faeces	n.d.	0% (0/35)		[109]
		Italy	Liver	n.d.	1.7% (1/60)	gt3	[110]
Moose	*Alces alces*	Sweden	Liver, kidney	n.d.	Liver: 1/6 kidney: 0/6	*P. alci*	[103]
		Poland	Serum	0% (0/1)	n.d.		[87]
		Sweden	Serum, faeces, liver	18.6% (43/231)	14.7% (34/231)	*P. alci*	[89]
		Sweden	Serum, faeces	14.0% (9/66)	15.1% (10/66)	*P. alci*	[92]
		Lithuania	Serum	n.d.	7.7% (1/13)		[107]
		Finland	Serum	9.1% (31/342)	0% (0/342)		[97]
		Norway	Serum	19.5% (32/164)	n.d.		[100]
Tufted deer	*Elaphodus cephalophus*	China	Faeces	n.d.	50.0% (4/8)	gt4	[102]
White tailed deer	*Odocoleicus virginianus*	Mexico	Serum	62.7% (89/142)	n.d.		[83]
		Canada	Serum	8.8% (18/205)	0% (0/205)		[84]
		Finland	Serum	1.4% (1/70)	0% (0/70)		[97]
Mule deer	*Odocoleicus hemionus*	Canada	Serum	4.5% (5/112)	0% (0/112)		[84]
Barren-ground caribou	*Rangifer tarandus* *groenlandicus*	Canada	Serum	1.7% (2/120)	0% (0/120)		[84]
Woodland caribou	*Rangifer tarandus* *caribou*	Canada	Serum	5.2% (5/97)	0% (0/97)		[84]
Eurasian tundra reindeer	*Rangifer tarandus tarandus*	Russia	Serum	12.0% (23/191)	0% (0/191)		[81]
		Norway	Serum	23.1% (43/186)	n.d.		[100]
		Norway	Serum	15.7% (81/516)	n.d.		[99]
Reeves’s muntjac	*Muntiacus reevesi*	China	Faeces	n.d.	50.0% (1/2)	gt4	[102]
		Japan	Serum	0% (0/1)	n.d.		[79]
Tatra chamois	*Rupicapra rupicapra tatrica*	Poland	Serum	0% (0/4)	n.d.		[87]
Chamois	*Rupricapra rupricapra*	Italy	Serum	1.2% (2/172)	0% (0/172)		[98]
		Italy	Serum, faeces	0% (0/13)	0% (0/13)		[64]
		Italy	Liver	n.d.	0% (0/4)		[108]
		Italy	Serum	5.1% (5/92)	n.d.		[101]
Alpine ibex	*Capra ibex*	Italy	Serum, faeces	6.3% (2/32)	0% (0/32)		[64]
Muskox	*Ovibos moschatus*	Norway	Serum	5.9% (6/102)	n.d.		[100]
European bison	*Bison bonasus*	Poland	Serum	0% (0/68)	n.d.		[87]
European muflon	*Ovis aries musimon*	Czech Republic	Faeces	n.d.	12.8% (5/39)		[104]
		Croatia	Blood, spleen, liver	n.d.	0% (0/12)		[51]
		Italy	Serum	2.0% (1/49)	n.d.		[101]
Persian gazelle	*Gazella subgutturosa*	Iran	Faeces	n.d.	6.0% (3/50)		[111]

^1^ n.d.: not determined.

## Data Availability

Not applicable.

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
