# Peer review of "Current Knowledge of Hepatitis E Virus (HEV) Epidemiology in Ruminants"

_pathogens, 2022, doi:10.3390/pathogens11101124_

Round 1
Reviewer 1 Report
Comments/suggestions are according to rows numeration
129: these are most probably pig farms; not "pork meat", unless they have a slaughtering line. Better explained in row 296
333: I would suggest considering the publication in: https://www.nature.com/articles/s41598-022-11208-6
relative to Middle East HEV in humans derived from camels consumption/contact
275; (serum +vity) this fact should induce to considerations relative to occupational risks of infection e.g slaughter plant workers (as partially discussed at row 348)
355-357; probably a study relative to slaughter plant workers could be suggested and eventually supported by a reference
Author Response
Reviewer 1 (R1)
R1.1 - 129: these are most probably pig farms; not "pork meat", unless they have a slaughtering line. Better explained in row 296
Reply to R1.1 - Many thanks to Referee 1 for this comment. The sentence “To explain the discrepancies observed between the European and Asiatic studies [31,49], it was speculated [50,51] that in the Asiatic countries small-sized farms with mixed production of pork meat and cow milk could generate a higher risk for HEV transmission to cattle from pigs.” was replaced with: “To explain the discrepancies observed between the European and Asiatic studies [33,52], it was speculated [53,54] that in the Asiatic countries small-sized farms with mixed animals could generate a higher risk for HEV transmission from pigs to cattle.”
R1.2 - 333: I would suggest considering the publication in: https://www.nature.com/articles/s41598-022-11208-6 relative to Middle East HEV in humans derived from camels consumption/contact
Reply to R1.2 - As suggested by Referee 1, the sentence “Human HEV infection through consumption of products of domestic ruminants has not been reported so far.” was replaced with the following: “Among ruminants, in addition to cluster cases directly linked to the consumption of raw deer meats, there are growing evidence on the role of domestic camels as zoonotic source of HEV gt7 infection [121], involved in a case of chronic hepatitis E after transplantation in a patient from the United Arab Emirates who regularly consumed camel milk and meat [20]. Human HEV infection associated with consumption of products from domestic cattle, goats and sheep has not been reported so far.” The following references were added in the revised manuscript:
[121] El-Kafrawy, S.A.; Hassan, A.M.; El-Daly, M.M.; Al-Hajri, M.; Farag, E.; Elnour, F.A.; Khan, A.; Tolah, A.M.; Alandijany, T.A.; Othman, N.A.; Memish, Z.A.; Corman, V.M.; Drosten, C.; Zumla, A.; Azhar, E.I. Genetic diversity of hepatitis E virus (HEV) in imported and domestic camels in Saudi Arabia. Sci Rep. 2022, 29, 12(1):7005. https://doi.org/10.1038/s41598-022-11208-6.
[20] Lee, G.H.; Tan, B.H.; Teo, E.C.; Lim, S.G.; Dan, Y.Y.; Wee, A.; Aw, P.P.; Zhu, Y.; Hibberd, M.L.; Tan, C.K.; Purdy, M.A.; Teo, C.G. Chronic Infection With Camelid Hepatitis E Virus in a Liver Transplant Recipient Who Regularly Consumes Camel Meat and Milk. Gastroenterology. 2016, 150(2):355-7.e3.https://doi.org/10.1053/j.gastro.2015.10.048.
R1.3 - 275; (serum +vity) this fact should induce to considerations relative to occupational risks of infection e.g slaughter plant workers (as partially discussed at row 348)
Reply to R1.3 – Following Referee 1 suggestions, the sentence “Serological surveillance in Portugal [70], has identified HEV antibodies in 29.3% of workers (shepherds and sheep milk cheesemakers) occupationally exposed to sheep or to ovine edible products but only in 16.1% of the control population, suggesting a potential occupational risk for HEV infection.” was modified as follows: “The potential occupational risk of HEV infection in setting with exposure to domestic ruminants has been assessed only in two studies, so far. In the serosurvey performed by Wu et al. [69], a seroprevalence of 57.7 % (15/26) was detected in butchers working in the same slaughterhouse in which sheep resulted positive for the presence of anti-HEV antibodies and for the presence of viral RNA in liver [69].” More recently, the occupational risk was investigated in shepherds and sheep milk cheesemakers-workers in a study in Portugal [73] revealing that the seroprevalence in workers was significantly higher (29.3%) when compared with the population controls (16.1%).” (See also response to R.1.4).
R1.4 - 355-357; probably a study relative to slaughter plant workers could be suggested and eventually supported by a reference
Reply to R1.4 - As suggested by the Referee 1, the following sentence was added in the conclusions section of the revised manuscript: “Furthermore, as several studies demonstrated the association between direct contact with swine and higher HEV seroprevalences in professionally exposed persons [130-132], additional serological surveillance for HEV in individuals with occupational exposure to ruminants, as slaughter plant workers, veterinarians, farmers and hunters, will be helpful in better understanding the role of ruminants as HEV host and the occupational risk linked to contact with them.” Also, the sentence “Furthermore, increasing controls for HEV in ruminant-derived production should be enacted.” was replaced by “Also, increasing devising surveillance plans to ascertain the viral hazards for humans associated with the consumption of ruminant-derived production should be enacted. For this purpose, current methods of nucleic acid extraction and purification, especially for food matrices, should be improved in order to avoid the decrease sensitivity of the molecular techniques employed.”. The following references were added in the revised manuscript:
[130] Krumbholz, A., Mohn, U., Lange, J., Motz, M., Wenzel, J.J., Jilg, W., Walther, M., Straube, E., Wutzler, P., Zell, R. Prevalence of hepatitis E virus-specific antibodies in humans with occupational exposure to pigs. Med Microbiol Immunol. 2012, 201 (2), 239–244. https://doi.org/10.1007/s00430-011-0210-5.
[131] Teixeira, J., Mesquita, J.R., Pereira, S.S., Oliveira, R.M., Abreu-Silva, J., Rodrigues, A., Myrmel, M., Stene-Johansen, K., Øverbø, J., Gonçalves, G., Nascimento, M.S. Prevalence of hepatitis E virus antibodies in workers occupationally exposed to swine in Portugal. Med Microbiol Immunol. 2017, 206 (1), 77–81. https://doi.org/10.1007/s00430-016-0484-8.
[132] Mrzljak, A., Balen, I., Barbic, L., Ilic, M., Vilibic-Cavlek, T. Hepatitis E virus in professionally exposed: A reason for concern?. World J Hepatol. 2021, 13 (7), 723–730. https://doi.org/10.4254/wjh.v13.i7.723.
Reviewer 2 Report
Dear Authors, the scientific literature about HEV in ruminants is wider than
I was expecting, so I appreciated the idea of ​​summarizing and comparing the content of existing publications to attempt an epidemiological assessment of general significance.
I have some comments and hints, hoping to help improve the paper:
- Introduction: I would suggest including an image representing a phylogenetic tree of Heperviridae, to help the comprehension of this complex family and putting in evidence the potential/zoonotic genotypes and species.
- I would also suggest discussing the potential limits of the diagnostic
sensitivity of some molecular techniques used in the cited studies, as
some methods are more or less targeted to specific virus genotypes.
- Line 101: “The first Identification of HEV RNA…” I suggest change to
“The first molecular genotyping”, as the first detection of HEV RNA was
referred to [26] in 2009.
- Line 103: the prevalence should be approximated to 8.8%, as in the table
- Line 107-108: I cannot find correspondence to the data 3.2% (3/92) [55]
in table n. 1
- Line 114-115: I would suggest citing the study published in 2011 before
the one published in 2015. About [49], it is important to clarify that
100% of positivity on milk samples is not a prevalence data, as only
selected animal (positive to some other tests) were considered. In the
text this concept is explained, but I would put a note in the table n. 1
at this propose, just to avoid misunderstanding (see also table n. 2,
reference [58]).
- Table 2, [22]: the seroprevalence showed is 42.0-67.0%, while at line 142
only 67.0% is commented.
- Table 2, [58] commented above, about 100% prevalence on milk samples. - Line 146: are there specific neutralizing antibodies not cross reacting
with some other genotypes?
- Line 156: subtype c or subtype C?
- Line 178: the data of [62] in table n.2 is 1.7% and in line 178 is 1.4%
- Line 193 and following: some data are not in agreement with table n.3:
Spain 2.6-2.1% (in the table) vs 1.9-2.1% (in the text) [44,63]; Nigeria
[64,68] seems to be missing; moreover, [64] in the table n. 2 is
referred to China, not to Nigeria; Portugal [70] and Egypt [38] seem
to be missing in the table; please re-check all the data.
- Line 207: “3/20 HEV-positive animals”: if I have well understood, I would
suggest to change to “3/20 viremic animals” - Wild ruminants: why did the Author decide not including a table dedicated
to them, as they did for the other species?
- Line 323-325: the finding of single positive animals (one red deer, one
chamois) may be due to a lack of specificity of the diagnostic
serological test. This point should be discussed, also considering that
diagnostic tests are rarely validated on wild species.
- Line 327: about unknown sources of infection, may the Authors consider
rodents as a potential risk?
- Line 341-344: a goat-to-human transmission was suggested. May the Author also consider a human-to-goat transmission?
- Line 350-352: I guess that forestry workers are in contact with wild
ruminants (deer), but also with wild boars. How to distinguish the
specific exposure risk?
Author Response
Reviewer 2 (R2)
Dear Authors, the scientific literature about HEV in ruminants is wider than was expecting, so I appreciated the idea of summarizing and comparing the content of existing publications to attempt an epidemiological assessment of general significance.
I have some comments and hints, hoping to help improve the paper:
R2.1 - Introduction: I would suggest including an image representing a phylogenetic tree of Heperviridae, to help the comprehension of this complex family and putting in evidence the potential/zoonotic genotypes and species.
Reply to R2.1 - In the revised manuscript, a figure (Fig. 1) showing the phylogenetic tree of the family Hepeviridae, was added. The following reference was added to the revised manuscript:
[21] Kumar S, Stecher G, Li M, Knyaz C, Tamura K. MEGA X: Molecular Evolutionary Genetics Analysis across Computing Plat-forms. Mol Biol Evol. 2018; 35 (6):1547-1549. Doi:10.1093/molbev/msy096.
R2.2 - I would also suggest discussing the potential limits of the diagnostic sensitivity of some molecular techniques used in the cited studies, as some methods are more or less targeted to specific virus genotypes.
Reply to R2.2 - As suggested by the Referee 2, in the Discussion section of the revised manuscript, the following sentences “.Hence, interpreting HEV serological data could be limited just by the inability of the molecular tools to date available to detect genetically highly divergent HEV strains as observed in the study of Geng et al. [31], where none of the antigen-positive goat or cattle samples resulted positive for HEV RNA. Also, as previously discussed by Yugo et al. [39], limits in viral molecular detection from ruminant samples due to the presence of amplification inhibitors, should be considered, especially during sample processing or nucleic acid extraction.”.” were added, also replying to Referee 4 (R4.1 and R4.5).
R2.3 - Line 101: “The first Identification of HEV RNA…” I suggest change to “The first molecular genotyping”, as the first detection of HEV RNA was referred to [26] in 2009.
Reply to R2.3 - In the revised manuscript “The first Identification of HEV RNA…” was replaced with “The first molecular genotyping”.
R2.4 - Line 103: the prevalence should be approximated to 8.8%, as in the table
Reply to R2.4 - In the revised manuscript the prevalence “8.7%” was approximated to “8.8%” as reported in the table.
R2.5 - Line 107-108: I cannot find correspondence to the data 3.2% (3/92) [55] in table n. 1
Reply to R2.5 - Many thanks to the Referee 2 for this observation. In the revised manuscript, the sentence “In a 2014 study, using specific molecular tools for gt4 HEV, viral RNA was found in 3.2% (3/92) of faecal samples collected from domesticated yaks (Bos grunniens) reared in Tibet Region for meat and milk [55]. The complete sequence analysis of one such strain revealed the highest nt identity to swine (99.1%) and human (93.8%) strains previously detected in China. Furthermore, HEV-positive yaks were found only in Qinghai, one of the two Provinces (Gansu and Qinghai) investigated.” was modified as follows: “In a 2014 study, using specific molecular tools for gt4 HEV, viral RNA was found in 1.8% (3/167) of faecal samples collected from domesticated yaks (Bos grunniens) reared in Tibet Region for meat and milk [58]. The complete sequence analysis of one such strain revealed the highest nt identity to swine (99.1%) and human (93.8%) strains previously detected in China. Furthermore, HEV-positive yaks (3/92, 3.3%) were found only in Qinghai, one of the two Provinces (Gansu and Qinghai) investigated.”
R2.6 - Line 114-115: I would suggest citing the study published in 2011 before the one published in 2015. About [49], it is important to clarify that 100% of positivity on milk samples is not a prevalence data, as only selected animal (positive to some other tests) was considered. In the text this concept is explained, but I would put a note in the table n. 1 at this propose, just to avoid misunderstanding (see also table n. 2, reference [58]).
Reply to R2.6– In the revised manuscript, the sentence “Subsequent molecular studies reported the detection of HEV RNA with rates of 37.1% (52/140) in stools of Holsteins cows in Yunnan Province (southwest China) and of 3.0% (8/254) in serum samples collected from yellow cattle (Bos taurus) of local breeds of the Shandong Province of Eastern China [31,49]” was replaced with “Subsequent molecular studies reported the detection of HEV RNA with rates of 3.0% (8/254) in serum samples collected from yellow cattle (Bos taurus) of local breeds of the Shandong Province of Eastern China and of 37.1% (52/140) in stools of Holsteins cows in Yunnan Province (southwest China) [33,52].”. Furthermore, in Table 1 of the revised manuscript, to better explain the significance of the value 100%, a footnote “* cohort of selected animals.” was added.
R2.7 - Table 2 [22]: the seroprevalence showed is 42.0-67.0%, while at line 142 only 67.0% is commented.
Reply to R2.7 - In Table 2 of the revised manuscript, the prevalence rate “42.0-67.0%” was replaced with the correct value “67.0%”.
R2.8 - Table 2, [58] commented above, about 100% prevalence on milk samples.
Reply to R2.8- as reported in the R2.6, to better explain the significance of the value 100%, a footnote “* cohort of selected animals.” was added at the end of Table 2.
R2.9 - Line 146: are there specific neutralizing antibodies not cross reacting with some other genotypes?
Reply to R2.9 - Although HEV strains are heterogeneous with highly genomic diversity, only one serotype of HEV exists (Primadharsini et al., 2019), as demonstrated in the study of Purcell et al. (2003) in which vaccination with a HEV gt1 capsid peptide had protected monkeys from challenge with gt1, gt2 or gt3 strains. Also, in the study of Emerson et al. (2006), broadly cross-reactive neutralizing antibodies to HEV were proved to be induced by infection with any one of the four major genotypes of the species Paslahepevirus balayani (previous Orthohepevirus A species). In the revised manuscript, the sentence “Interestingly, in a study conducted in Virginia (USA) IgG antibodies anti-HEV were revealed in 16.0% (13/80) of goats, the majority of which had neutralizing antibodies to the human gt1 HEV strain Sar-55 [56]. It was also observed seroconversion in 7 of 11 kids, monitored from birth until 14 weeks of age, although HEV RNA was not detected neither in the faecal nor in the serum samples [56]. Attempts to infect goats experimentally with three different HEV genotypes (gt1-Sar-55, gt3-Meng and gt4-TW6196E) were unsuccessful [56].” was modified as follows: “Interestingly, in a study conducted in Virginia (USA) IgG antibodies anti-HEV were found in 16.0% (13/80) of goats tested, also demonstrating the presence of neutralizing antibodies to HEV in selected IgG anti-HEV positive goat sera [59]. However, attempts to infect goats experimentally with three different HEV genotypes (gt1-Sar-55, gt3-Meng and gt4-TW6196E) were unsuccessful [59].” Furthermore, the sentence “It was also observed seroconversion in 7 of 11 kids, monitored from birth until 14 weeks of age, although HEV RNA was not detected neither in the faecal nor in the serum samples [56].” was moved to the Discussion section with little modifications “A similar finding had been previously described for goats in the study of Sanford et al. [59], in which seroconversion was observed in 7 of 11 kids, monitored from birth until 14 weeks of age, although HEV RNA was not detected neither in the faecal nor in the serum samples [59].”
R2.10 - Line 156: subtype c or subtype C?
Reply to R2.10 – In the revised version, lowercase letters were used for subtypes throughout the manuscript.
R2.11 - Line 178: the data of [62] in table n.2 is and in line 178 is 1.4%
Reply to R2.11 - many thanks to the Referee 2. In the Table 2 of the revised manuscript, “1.7%” value was replaced with the correct value of “1.4%”.
R2.12 - Line 193 and following: some data are not in agreement with table n.3: Spain 2.6-2.1% (in the table) vs 1.9-2.1% (in the text) [44,63]; Nigeria [64,68] seems to be missing; moreover, [64] in the table n. 2 is referred to China, not to Nigeria; Portugal [70] and Egypt [38] seem to be missing in the table; please re-check all the data.
Reply to R2.12 – many thanks to the Referee 2. All data reported in the Tables 2 and 3, were checked and corrected in the revised manuscript.
R2.13 - Line 207: “3/20 HEV-positive animals”: if I have well understood, I would suggest to change to “3/20 viremic animals”.
Reply to R2.13 – In the revised manuscript the sentence “Interestingly, 3/20 HEV-positive animals were also shedding HEV in the faeces.” was modified as follows: “Interestingly, 3/20 HEV-positive animals were also viremic.”.
R2.14 - Wild ruminants: why did the Author decide not including a table dedicated to them, as they did for the other species?
Reply to R2.14- as suggested by the Referee 2, a new table (Table 4), reporting serological and molecular prevalence data of HEV in wild ruminants, was added in the revised manuscript.
R2.15 - Line 323-325: the finding of single positive animals (one red deer, one chamois) may be due to a lack of specificity of the diagnostic serological test. This point should be discussed, also considering that diagnostic tests are rarely validated on wild species.
Reply to R2.15 - In the revised manuscript the original sentence “In these cases, direct or indirect contact with wild boar is unlikely to account for exposure to HEV and there could be unknown sources of HEV infection for wild ruminants [95].” was modified as follows: “In these cases, direct or indirect contact with wild boar is unlikely to account for exposure to HEV. Even though a lack of specificity of the serological assay employed could be considered, as it was also observed in the study of Rutjies et al. [86], in which it was suspected an overestimated prevalence for deer species, on the other hand it cannot be excluded that there could be unknown sources of HEV infection for wild ruminants [98].”
R2.16 - Line 327: about unknown sources of infection, may the Authors consider rodents as a potential risk?
Reply to R2.16 - many thanks to the Referee 2 for this observation. In the revised manuscript, the following paragraph was added in the Discussion section: “Rinaldo et al. [99], suggest the possibility that hares and rats could potentially have been involved in the transmission of HEV to Norwegian semi-domesticated reindeer in an area in which wild boars have never been observed [99]. The role of rodents in HEV infections have been repeatedly investigated, mostly on pig farms due to their potential abundance on farms. In a review, analysing infection dynamics and persistence of hepatitis E virus on pig farms, was discussed that the low prevalence of HEV gt3 in rodents around farms and the detection predominantly in intestines, supports the argument that rodents are only accidental hosts of HEV gt3. Likely, rodents may be considered a potential risk that mechanically contribute to spread of infected faecal material and as such contribute to environmental contamination [115].”. The following reference was added in the revised manuscript:
[115] Meester, M., Tobias, T.J., Bouwknegt, M., Kusters, N.E., Stegeman, J.A., van der Poel, W. Infection dynamics and persistence of hepatitis E virus on pig farms - a review. Porcine Health Manag. 2021, 7 (1), 16. https://doi.org/10.1186/s40813-021-00189-z.
R2.17 - Line 341-344: a goat-to-human transmission was suggested. May the Author also consider a human-to-goat transmission?
Reply to R2.17-In the study of El-Mokhtar et al., (2020) a human-to-goat transmission was not considered. In the revised manuscript the sentence “The risk of HEV transmission to Egyptian goat owners was assessed, and none of the households owning seronegative goats had HEV markers, whilst 80% of households owning seropositive goats also tested positive for anti-HEV IgG, suggesting a possible goat-to-human transmission [60]. Likewise, higher anti-HEV IgG seroprevalence was reported in cattle farmers than in other inhabitants of Lao village [32].” was modified as follows “The risk of HEV transmission to Egyptian goat owners was assessed, and none of the households owning seronegative goats had HEV markers, whilst 80% of households owning seropositive goats also tested positive for anti-HEV IgG, suggesting a potential interaction between goats and households. In detail, in this study, a possible goat-to-human transmission was considered [63], whilst a possible transmission human-to-goat, although could not be completely excluded, was not discussed. However, in a survey conducted in Lao village [34], higher anti-HEV IgG seroprevalence has been reported in cattle farmers compared to other villagers [34], thus supporting the possible animal-to-human transmission.”.
R2.18 - Line 350-352: I guess that forestry workers are in contact with wild ruminants (deer), but also with wild boars. How to distinguish the specific exposure risk?
Reply to R2.18 - It is difficult to distinguish the specific exposure risk of forestry workers in contact with several species of wild animals. In the available and cited studies in the present review, risks exposure considered were referred to wild species, considering mainly wild boars but also deer. Accordingly, in the revised manuscript the sentence “Similarly, several serological studies suggest that persons with occupational contact with wild animals (deer) have a higher seroprevalence than the related general population [117-121]. The exposure risk to HEV infection in forestry workers was documented in two studies performed in Germany and France. Higher HEV seroprevalences were found in these groups than in the control groups [117,118].” was modified as follows: “Similarly, several serological studies suggest that populations workers having occupational contact with wild animals, mainly represented by wild boars and deer, have a higher seroprevalence than the related general population [123-129]. The exposure risk to HEV infection in forestry workers was documented in studies performed in Germany and France, higher HEV seroprevalences were found in these groups than in the control groups [123-125].” The following references were added in the revised manuscript:
[124] Carpentier, A., Chaussade, H., Rigaud, E., Rodriguez, J., Berthault, C., Boué, F., Tognon, M., Touzé, A., Garcia-Bonnet, N., Choutet, P., Coursaget, P. High hepatitis E virus seroprevalence in forestry workers and in wild boars in France. J Clin Microbiol. 2012, 50 (9), 2888–2893. https://doi.org/10.1128/JCM.00989-12.
[127] Hartl, J., Otto, B., Madden, R.G., Webb, G., Woolson, K.L., Kriston, L., Vettorazzi, E., Lohse, A.W., Dalton, H.R., Pischke, S. Hepatitis E Seroprevalence in Europe: A Meta-Analysis. Viruses. 2016 8 (8), 211. https://doi.org/10.3390/v8080211.
Reviewer 3 Report
In the current review article Di Profio et al summarizes the accumulated knowledge on HEV infection in Ruminants. The subject is very interesting and relevant and the study is well written. The angel on domestic bovine cohorts is especially interesting and could have an impact on practice in the diary and beef industry. However, the review would benefit of a work-up to make the presentations of the studies more accessible and uncover more general tendencies.
Specific comments:
1. Include world wide maps of the HEV prevalence in the various species of ruminents.
2. On HEV prevalence in cattle summarize differences between China and Eoropean countries please visualize in a bar graph with statistical comparison.
3. Table 2, please edit the document so that the whole table is on one page.
4. Lines 147-149. This sentence is somewhat disconnected to the rest of the text. Please frame the sentence or remove.
5. Line 200. Please include table on frequencies of HEV in various tissues for the different ruminant species that the review comprises.
6. Table 3. Please edit the document so that the whole table is on one page.
7. Please include table of HEV prevalence in wild ruminents in various sample materials and tissues.
Author Response
Reviewer 3 (R3)
In the current review article Di Profio et al summarizes the accumulated knowledge on HEV infection in Ruminants. The subject is very interesting and relevant and the study is well written. The angel on domestic bovine cohorts is especially interesting and could have an impact on practice in the diary and beef industry. However, the review would benefit of a work-up to make the presentations of the studies more accessible and uncover more general tendencies.
Specific comments:
R3.1 - Include worldwide maps of the HEV prevalence in the various species of ruminants.
Reply to R3.1 – as suggested by Referee 3, in the revised manuscript a figure (Figure 2A and 2B) showing the global distribution of HEV molecular studies performed in wild (A) and domestic ruminants (B), was added.
R3.2 - On HEV prevalence in cattle summarize differences between China and European countries please visualize in a bar graph with statistical comparison.
Reply to R3.2 – in the revised manuscript a figure (bar graph) showing HEV seroprevalence rates detected in cattle in European (Figure 3A) and Chinese surveys (Figure 3B), was added.
R3.3 - Table 2, please edit the document so that the whole table is on one page.
Reply to R3.3 – in the revised manuscript the whole Table 2 was adapted in a unique page.
R3.4 - Lines 147-149. This sentence is somewhat disconnected to the rest of the text. Please frame the sentence or remove.
Reply to R3.4 – In the revised manuscript, the sentence “It was also observed seroconversion in 7 of 11 kids, monitored from birth until 14 weeks of age, although HEV RNA was not detected neither in the faecal nor in the serum samples [56].” was moved to the “Discussion” section of the revised manuscript and modified as follows “A similar finding had been previously described for goats in the study of Sanford et al. [59], in which seroconversion was observed in 7 of 11 kids, monitored from birth until 14 weeks of age, although HEV RNA was not detected neither in the faecal nor in the serum samples [59].”
R3.5 - Line 200. Please include table on frequencies of HEV in various tissues for the different ruminant species that the review comprises.
Reply to R3.5 – In the revised manuscript, a detailed description of tissue samples in which HEV RNA was identified, was reported in all the table (Table 1, 2, 3 and 4) to ruminant species investigated.
R3.6 - Table 3. Please edit the document so that the whole table is on one page.
Reply to R3.6 - In the revised manuscript the whole Table 3 was adapted in a unique page.
R3.7 - Please include table of HEV prevalence in wild ruminants in various sample materials and tissues.
Reply to R3.7 – In the revised manuscript a new table (Table 4) reporting data on wild ruminants including tissue samples was added.
Reviewer 4 Report
It is an interesting and hop topic which discussed a controversial point. While some studies confirmed that ruminants are a reservoir and source of infection for HEV, others either did not detect HEV markers in the ruminants/ products or hypothesized that they are HEV-like agents circulating in some ruminants such as cows and goats.
Some points need to be covered by the authors to improve the quality of the review.
Major points
1- Methods of nucleic acid extraction and purification from the ruminant products could affect the detection rate of HEV on ruminants and therefore the epidemiology data
2- I found one commentary discussed the same topic https://doi.org/10.2217/fvl-2021-0188, the authors should discuss this one in terms or updates.
3- The authors should include new direction and future perspective for this topic. Also future expectations.
4-Unless I missed, the authors should discussed the circulation of HEV in ruminants in terms of different genotypes, subtypes, geographic locations, habits, mixing with other animals.
5- The authors should also discuss why some reports found anti-HEV Igs in the ruminants, but could not found HEV RNA or Ag in the same animals.
Author Response
Reviewer 4 (R4)
It is an interesting and hop topic which discussed a controversial point. While some studies confirmed that ruminants are a reservoir and source of infection for HEV, others either did not detect HEV markers in the ruminants/ products or hypothesized that they are HEV-like agents circulating in some ruminants such as cows and goats.
Some points need to be covered by the authors to improve the quality of the review.
Major points
R4.1 - Methods of nucleic acid extraction and purification from the ruminant products could affect the detection rate of HEV on ruminants and therefore the epidemiology data
Reply to R4.1 – In the discussion section of the revised manuscript, the following sentence was added “Also, as previously discussed by Yugo et al. [39], limits in viral molecular detection from ruminant samples due to the presence of amplification inhibitors, should be considered, especially during sample processing or nucleic acid extraction.”(See also response to R2.2)
R4.2 - I found one commentary discussed the same topic https://doi.org/10.2217/fvl-2021-0188, the authors should discuss this one in terms or updates.
Reply to R4.2- many thanks to the Referee 4. In the cited commentary, the Authors discus the available studies supporting or opposing the role of ruminants and their products as potential sources of human hepatitis E virus infection. In the “Discussion” section of the revised manuscript, the following sentence was added: “As Sayed and El-Mokhtar [122] previously discussed, ingestion of ruminants raw milk products and undercooked liver should be considered a potential risk factor for HEV transmission to humans.”. Furthermore, taking advantages by the reading of the commentary recommended by the Referee, the sentences “Especially in rural communities, households typically own a small number of cows, sheep and/or goats raised for milk and meat production. These animals, in particular small ruminants, are social and docile, and this may favour close contacts and interactions, increasing the risks of zoonotic infection.” were modified as follows: “Furthermore, especially in rural communities, where households typically own a small number of cows, sheep and/or goats raised for milk and meat production intended for personal consumption, there could be a high exposure to animal contact [122]. Indeed, these species, in particular small ruminants, are social and docile, and this may favor close contacts and interactions, increasing the risks of zoonotic infection.”. The following reference was added:
[122] Sayed, I.M.; El-Mokhtar, M.A. Are ruminants and their products potential sources of human hepatitis E virus infection?. Future Virology. 2021, 16 (12), 785-789. https://doi.org/10.2217/fvl-2021-0188.
R4.3 - The authors should include new direction and future perspective for this topic. Also future expectations.
Reply to R4.3 – In the Discussion section of the revised manuscript, the following sentence was added: “Screening to identify the risk of HEV infection with the consumption of these ruminant products should be routinely performed, especially for products sold in local markets, but also for other edible ruminant organs, including intestine, kidney, brain and spleen [122].”Also, in the Conclusions section this paragraph was added: “Furthermore, as several studies demonstrated the association between direct contact with swine and higher HEV seroprevalences in professionally exposed persons [130-132], additional serological surveillance for HEV in individuals with occupational exposure to ruminants, as slaughter plant workers, veterinarians, farmers and hunters, will be helpful in better understanding the role of ruminants as HEV host and the occupational risk linked to contact with them. Also, increasing devising surveillance plans to ascertain the viral hazards for humans associated with the consumption of ruminant-derived production should be enacted. For this purpose, current methods of nucleic acid extraction and purification, especially for food matrices, should be improved in order to avoid the decrease sensitivity of the molecular techniques employed.”. (See also answer to R1.4). The following reference was added in the revised manuscript:
[130] Krumbholz, A., Mohn, U., Lange, J., Motz, M., Wenzel, J.J., Jilg, W., Walther, M., Straube, E., Wutzler, P., Zell, R. Prevalence of hepatitis E virus-specific antibodies in humans with occupational exposure to pigs. Med Microbiol Immunol. 2012, 201 (2), 239–244. https://doi.org/10.1007/s00430-011-0210-5.
[131] Teixeira, J., Mesquita, J.R., Pereira, S.S., Oliveira, R.M., Abreu-Silva, J., Rodrigues, A., Myrmel, M., Stene-Johansen, K., Øverbø, J., Gonçalves, G., Nascimento, M.S. Prevalence of hepatitis E virus antibodies in workers occupationally exposed to swine in Portugal. Med Microbiol Immunol. 2017, 206 (1), 77–81. https://doi.org/10.1007/s00430-016-0484-8.
[132] Mrzljak, A., Balen, I., Barbic, L., Ilic, M., Vilibic-Cavlek, T. Hepatitis E virus in professionally exposed: A reason for concern?. World J Hepatol. 2021, 13 (7), 723–730. https://doi.org/10.4254/wjh.v13.i7.723.
R4.4 - Unless I missed, the authors should discussed the circulation of HEV in ruminants in terms of different genotypes, subtypes, geographic locations, habits, mixing with other animals.
Reply to R4.4 - As suggested by the Referee 4, the paragraphs from line 292 to 306 of the original manuscript was modified and integrated as follows: “This hypothesis is also supported by the recurrent detection in cattle, sheep, goats and several wild ruminants of gt3 and gt4 HEV strains genetically highly related to HEVs identified in pigs or wild boars in the same geographical area [33,50,58,60,61,74]. In domestic ruminants, HEV prevalence appears higher in rural areas with traditional mixed farming systems, consisting in family-based small-sized farms hosting pigs and other domestic animals, a favorable epidemiological picture that may foster inter-species interactions [26,31,49,58,60]. In a recent study in Burkina Faso [45], cattle seropositivity was significantly associated with the presence of pigs in the same farm, suggesting the role of swine as risk factor [45]. However, that is not always the case, since Geng et al. [55] didn’t detect HEV in the faeces and milk of cows reared in mixed farms or neighboring farms with pigs in Hebei province of China [55] as well in Belgium where viral RNA was not detected in cow fecal and milk samples even collected in a mixed farm, in which HEV infection of swine had been demonstrated [54]. In addition, Gt3 HEV strains were detected in cows and goats of Assiut village, Egypt, where pig farms are not common, and even in the area housing pig farms, mixing of pigs and cows is uncommon due to religious beliefs [44,63]. Accordingly, HEV transmission to domestic ruminants could be due to several factors including husbandry practices, type and intensity of inter-species contacts as well as hygiene on the farms and handling/management of HEV contaminated manure produced by different animal species [44,45]. In a survey performed in the Czech Republic [65], it was speculated that pastures contaminated by HEV positive wild animals (wild boars, red deer, roe deer and mouflons) could be the source of infection for small ruminants in which HEV RNA was detected in raw milk samples [65]. In a study conducted in Jordan [35] several farm management practices were significantly associated with HEV seroprevalence at the farm-level. Large and small dairy ruminant farms that reported infrequent cleaning of feeder stations and infrequent general farm cleaning or mixing small ruminants (sheep and goats) together in the same flock had greater odds of HEV seroprevalence.”.
R4.5 - The authors should also discuss why some reports found anti-HEV Igs in the ruminants, but could not found HEV RNA or Ag in the same animals.
Reply to R4.5 – The following paragraph was added in the Discussion section of the revised manuscript: “In a prospective study conducted in a seropositive dairy herd, monitoring of newborn calves from birth to six months of age, revealed seroconversion to IgG anti-HEV. However, despite the several attempts either by using broad- spectrum RT-PCR assays or next generation sequencing approach, viral RNA was not detected. It was hypothesized that cattle may be susceptible to antigenically-related strains, still genetically uncharacterized, inducing cross-reactive HEV antibodies [39]. A similar finding had been previously described for goats in the study of Sanford et al. [59], in which seroconversion was observed in 7 of 11 kids, monitored from birth until 14 weeks of age, although HEV RNA was not detected neither in the faecal nor in the serum samples [59]. Hence, interpreting HEV serological data could be limited just by the inability of the molecular tools to date available to detect genetically highly divergent HEV strains as observed in the study of Geng et al. [31], where none of the antigen-positive goat or cattle samples resulted positive for HEV RNA. Also, as previously discussed by Yugo et al. [39] limits in viral molecular detection from ruminant samples due to the presence of amplification inhibitors, should be considered, especially during sample processing or nucleic acid extraction.”(See also response to R2.2)
Round 2
Reviewer 2 Report
No additional comments
Reviewer 4 Report
The authors addressed most of my comments.